# Does the 11-year solar cycle affect lake and river ice phenology?

**Daniel F. Schmidt** [1,2]*, **Kevin M. Grise** [2], **Michael L. Pace** [2]

**1** Department of Mathematics, Liberty University, Lynchburg, Virginia, United States of America,
**2** Department of Environmental Sciences, University of Virginia, Charlottesville, Virginia, United States of America

\* dfschmidt@liberty.edu

## Abstract

Records of ice-on and ice-off dates are available for lakes and rivers across the Northern Hemisphere spanning decades and in some cases centuries. This data provides an opportunity to investigate the climatic processes that may control ice phenology. Previous studies have reported a trend toward shorter ice-covered seasons with global warming, as well as links between ice phenology and several modes of natural climate variability such as the North Atlantic Oscillation, the Pacific-North American Pattern, the El Niño-Southern Oscillation, the Pacific Decadal Oscillation, and the Atlantic Multidecadal Oscillation. The 11-year sunspot cycle has also been proposed as a driver of ice phenology, which is somewhat surprising given that this cycle's strongest impacts are in the stratosphere. In this study, we use a large data set of lakes and rivers across the Northern Hemisphere to test this potential link. We find little or no connection between the sunspot cycle and either ice-on or ice-off dates. We conclude that while many well-known climate cycles do impact ice phenology, we are able to rule out any strong impact of the solar cycle.

**Data Availability Statement:** Ice data are available from the NSIDC (https://nsidc.org/data/lake_river_ice/). Sunspot data are available from WDC-SILSO (https://www.sidc.be/SILSO/datafiles). All other data are available from ECMWF (https://www.

## Introduction

The phenology of ice on inland water bodies affects the annual cycles of mixing and stratification with resulting impacts on ecosystem processes. Recent studies emphasize the importance of winter conditions and the activity of various ecological communities (e.g., plankton growth, biogeochemical cycling) during this time of the year [1, 2]. Ice phenology is changing as climate change causes water bodies to freeze later and thaw earlier [3, 4]. Earlier ice-off, in turn, leads to more rapid warming of water temperatures [5]. Hence, inland water ecosystems are also changing because the period of ice cover is declining, and in some cases, transitions are occurring to annual cycles that are ice free [6].

In addition to its importance for aquatic ecosystems, lake and river ice is an interesting subject from a climate science perspective because ice phenology data is often available for longer time periods than other climate data. The most detailed climate data is available for the duration of the satellite era (1979—present), and while some data does exist for considerably earlier times, its interpretation is complicated by differences in instrumentation and sparser coverage.

ecmwf.int/en/forecasts/dataset/ecmwf-reanalysis-v5).

**Funding:** MLP received funding from National Science Foundation grant DEB-1754712. The funders had no role in study design, data collection and analysis, decision to publish, or preparation of the manuscript.

**Competing interests:** The authors have declared that no competing interests exist.

On the other hand, freezing dates and ice breakup dates ("ice-on" and "ice-off" respectively) are easy to observe without sophisticated instruments and have been recorded at various locations across the Northern Hemisphere for decades to centuries. This data therefore provides an inviting opportunity to study the climatic drivers of lake and river ice phenology.

Most obviously, anthropogenic climate change over the last several decades to a century should have driven a trend toward shorter ice-covered seasons, consistent with warming global surface temperatures. Numerous studies document this trend [4, 7–18], along with a trend toward warmer water temperatures and associated changes in freshwater ecosystems [19–22]. Trends in ice data are typically highly significant, and in most (but not all) cases ice-on is trending later and ice-off is trending earlier, as would be expected with global warming. However, these trends occur against a strong background of year-to-year variability, with ice-on and ice-off dates frequently deviating from the trendline by as much as 10–15 days (Fig 1).

This remaining variability may be explained in part by natural modes of climate variability. Examples would include the North Atlantic Oscillation (NAO), Pacific-North America Pattern (PNA), El Niño-Southern Oscillation (ENSO), Pacific Decadal Oscillation (PDO) and Atlantic Multidecadal Oscillation (AMO). In contrast to centennial-scale anthropogenic climate change, these processes are internal to the climate system and operate on timescales ranging from weeks to decades. These processes can influence near-surface air temperatures in the Northern Hemisphere, which in turn would presumably change the timing of ice-on and/or ice-off. Indeed, several studies have found links between lake ice phenology and some of these natural modes of variability [7, 10, 14, 17, 23–27], though generally the influence of the higher-frequency modes (NAO, PNA, and ENSO) is clearer [16].

A third possible driver of ice phenology is solar variability. In contrast to the previous two categories, this driver is neither anthropogenic nor internal to the climate system, but represents a natural, cyclic change in solar forcing. Specifically, observers have for centuries noted a clearly defined 11-year cycle in the number of sunspots visible on the disk of the sun. This cycle is so clear that it can be seen even with relatively rudimentary instruments and temporally sparse data. Thus, the available time series is centuries long, and the cycle is clearly evident in the time series data (top panel, Fig 2) as well as its power spectrum generated using a fast Fourier transform (bottom panel, Fig 2). (See "Data and Methods" section below for the source of the sunspot data).

More recently, astronomers have discovered that the number of sunspots is positively correlated with solar luminosity, meaning that the so-called "solar constant" also has an 11-year periodicity, though the amplitude is quite small (around 0.1% of total solar luminosity; [28, 29]). Given that the entire climate system is driven by solar radiation, it is natural to wonder whether this 11-year cycle in solar luminosity—even a small one—has some noticeable impact on the climate system. This question has indeed attracted considerable attention [30–35].

Work in this field has suggested that the impact of the solar cycle is strongest in the stratosphere [29]. This is because, while the total solar luminosity changes by only about 0.1% with the solar cycle, the change in ultraviolet luminosity is orders of magnitude larger [28, 29, 36]. Much of this ultraviolet light is absorbed by stratospheric ozone and thus produces a reasonably strong signal in stratospheric temperatures—perhaps as much as 3 K difference between solar maximum and minimum depending on which part of the stratosphere one considers [37]. The correlation between near-surface temperature and the solar cycle, on the other hand, is generally fairly weak, though it can differ noticeably from zero in some locations [38]. Impacts on surface weather have also turned out to be quite weak, if detectable at all [28, 39, 40]. It is therefore not immediately clear whether the solar cycle should have any discernible influence on ice phenology, though several previous studies have reported a possible link, at least for certain specific locations [17, 26, 41]. If such a link were real (and if it were reasonably

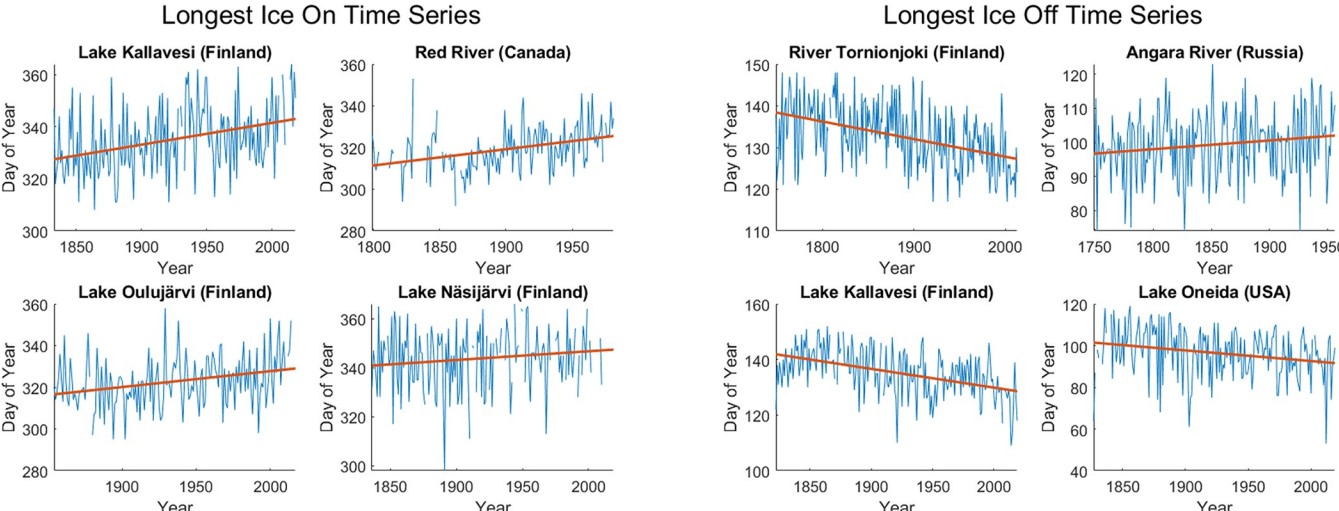

**Fig 1.** The four longest ice-on time series (left panels) and the four longest ice-off time series (right panels). Red lines are trendlines generated using least-squares linear regressions. Note that the axes differ between panels. Note also that the longest ice-on timeseries and the longest ice-off time series do not always occur at the same locations.

strong at some locations), it would provide an interesting opportunity for forecasting ice phenology years in advance.

In this study we consider six questions using six different methods to assess possible connections between ice phenology and the solar cycle: (1) based on the sensitivity of temperature to the solar cycle and the sensitivity of ice to temperature, should we expect the connection to be strong enough to be measurable? (2) Is there any detectable correlation between the 11-year sunspot cycle and ice phenology at individual locations? (3) Is there a lagged correlation between the 11-year sunspot cycle and ice phenology at individual locations? (4) Does the Fourier transform of ice data show any 11-year cycles? (5) Is there any spatial coherence in the correlations at individual locations? (6) Does the spatial pattern of ice-vs-sunspot correlations match the spatial pattern of temperature-vs-sunspot correlations? (For example, do locations that are warmest at solar maximum also have later ice-on dates at solar maximum?).

## Data and methods

We use ice-on and ice-off data from the Global Lake and River Ice Phenology Database maintained by the National Snow and Ice Data Center (NSIDC) [42]. The ice-on and ice-off dates for a lake or river are defined as the first day that the body of water is completely ice covered, and the first day that it is not completely ice covered, respectively. We convert all dates to day-of-year or "ordinal date" format—in which February 15, for example, would be day 46. To avoid aliasing of the climate change signal, we use de-trended values for all variables other than the sunspot number. (To de-trend a variable, we calculate the least-squares linear regression of the variable onto time and then subtract this regression from the original variable.) We exclude years in which a lake or river did not freeze, but these are rare (less than 1% of the data set). We also treat an ice-on date of January 1, for example, as if it was day 366 of the previous year (367 if the previous year was a leap year), so that a correlation or regression analysis will be able to treat this as later than an ice-on date of December 31. (Cases like this are also rare.) In contrast, missing data—representing years in which nothing was recorded—are common. We exclude missing years rather than attempting to interpolate. The full data set lists 868 lakes and rivers. However, the length of the available time series varies dramatically (see S1 Table),

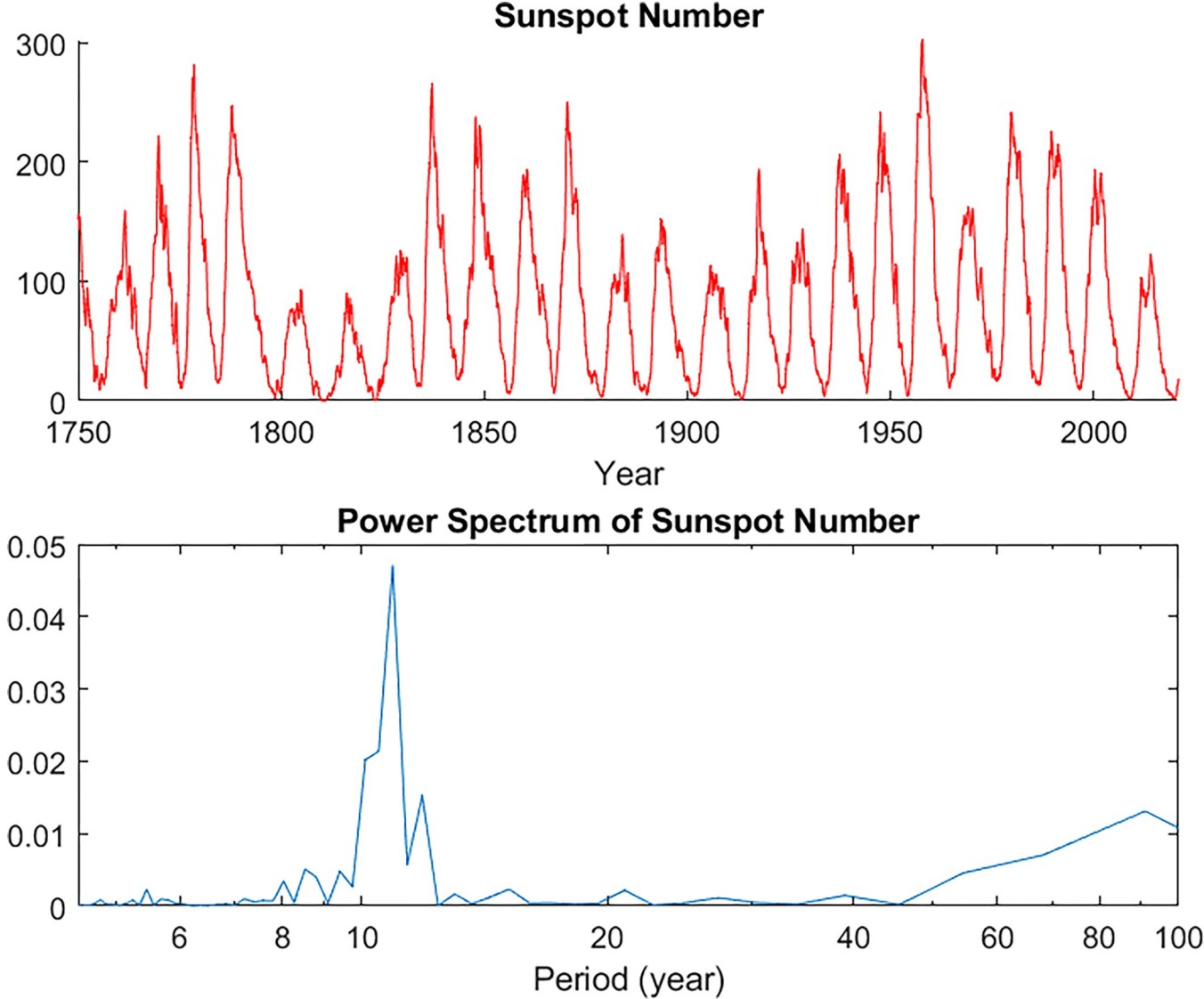

**Fig 2.** Time series of sunspot number (top panel) and the Fourier transform of the same data (bottom panel). Note that both show a very clear 11-year cycle and a weaker 90-year cycle. Note also that while Fourier transforms are normally displayed as functions of frequency, we have used period as the horizontal axis here. The vertical axis on the second panel has been normalized so that the area under the curve would be 1 if it was plotted on a linear horizontal axis.

and we only use bodies of water with at least 22 years of data (which may be non-consecutive due to missing years), corresponding to about two full repetitions of the solar cycle. This leaves 355 locations with enough ice-on data to analyze, and 456 for ice-off. (See S2 Table for a summary of data availability by decade.)

To track the solar cycle, we use monthly-mean values of the Wolf Sunspot Number from January 1818 to August 2022. This dataset is maintained by WDC-SILSO, Royal Observatory of Belgium, Brussels.

We also use monthly-mean 2-meter air temperature and monthly total precipitation from the European Centre for Medium-Range Weather Forecasts (ECMWF) fifth-generation reanalysis data set (ERA5; [43]) for the years 1940–2022 at 1˚ resolution on a latitude-longitude grid.

For this analysis, we often average both temperatures (or precipitation totals) and sunspot numbers over two seasons, the November-December-January (NDJ) season, and the March-April-May (MAM) season. We choose those months so that the sunspot numbers are taken from a time of year that corresponds reasonably well to the freeze or melt dates for most lakes and rivers in the data set. However, we also note that the solar cycle changes so slowly that the exact choice of months used in the average is unlikely to change the results.

## Results

### Sensitivity analysis

Before attempting to detect a correlation between ice phenology and the solar cycle, we ask whether we should even expect such a correlation. Specifically, if we compute the sensitivity of ice-on and ice-off dates to temperature, and the sensitivity of temperature to the solar cycle, we should be able to estimate the sensitivity of ice-on or ice-off to the solar cycle. Suppose the ice-on or ice-off date for the lake or river at location $k$ is labeled by $D_k$, the 2-meter air temperature at that location (at 1° resolution) by $T_k$, and the sunspot number by $S$. Then by the chain rule for derivatives,

$$\frac{dD_k}{dS} = \left(\frac{dD_k}{dT_k}\right)\left(\frac{dT_k}{dS}\right)$$

This assumes that no other variables are involved, and that the dependence of $D_k$ on $S$ is mediated by $T_k$ alone. If this is not the case, then a more complicated form of the chain rule may be needed, but this simple version should suffice for an order-of-magnitude estimate. To make the result easier to interpret, we multiply by 150, which is roughly the difference in sunspot number between a typical solar maximum and solar minimum. (However, note that while sunspot number at solar minimum are fairly consistent, those at solar maximum can differ considerably from one cycle to the next—see Fig 2). We perform this analysis only at locations in which at least 22 years of ice data are available. Note that these may not be the same locations for ice-off as for ice-on.

We find that on average, ice-on is delayed by 1.5 days for each +1°C temperature anomaly in the NDJ season, whereas ice-off occurs 3.8 days early for each +1°C temperature anomaly in the MAM season. The sensitivity of temperature to the solar cycle differs by location—and even the sign is not consistent—but its absolute value averages about 0.24°C difference for a sunspot number difference of 150 (not shown).

We multiply these two regressions at each location (which will not necessarily give the same answer as multiplying the average values given above). We find that ice-on should change by about 0.41 days between solar maximum and solar minimum, and ice-off should change by about 0.78 days. The signs differ depending on location (not shown). Thus, the expected connection between ice phenology and the solar cycle is small and would likely be difficult—though perhaps not impossible—to detect. Notably though, the connection might be easier to detect for ice-off, and indeed this is reflected in our subsequent analyses below.

It is possible that the solar cycle affects ice phenology via some variable other than temperature, so we repeat the above analysis with temperature replaced by precipitation. We then find that ice-on should change by about 0.19 days between solar maximum and solar minimum, and ice-off should change by about 0.21 days, when the effect is mediated by precipitation alone. Once again, the signs differ by location. Thus, we conclude that the effect of the solar cycle—if there is any effect at all—is likely to be mediated more by temperature than by precipitation.

## Individual correlations

The next method is simple: we check the correlations (specifically, Pearson correlation coefficients) between ice phenology and the solar cycle for individual lakes and rivers. More specifically, for each body of water, we calculate the correlation between the timeseries of ice-on dates and the timeseries of sunspot numbers, using the NDJ average for the latter so that each timeseries (temperature and ice-on) has one entry per year. Separately, we calculate the timeseries of ice-off dates with the timeseries of MAM-averaged sunspot numbers.

The top left panels of Figs 3 and 4 show these correlations on a map for both ice-on (Fig 3) and ice-off (Fig 4) with red and blue dots indicating positive and negative correlations, respectively, and the size of the dots indicating the magnitude of the correlation. In both panels, the positive and negative correlations are scattered throughout the map with no clear spatial organization. Furthermore, very few of these correlations are statistically significant. For ice-on, there are only two locations with correlations that are significant at the 0.05 level (Gull Lake and Athabasca River with 29 and 24 years of data, respectively), and for ice-off, there is only one (Hillsborough River, 24 years of data). However, out of 355 locations with at least 22 years of ice-on data (456 for ice-off), a few statistically significant results are to be expected by chance, even if there is no real causal relationship between the solar cycle and ice, and it is noteworthy that these few significant results occur for locations with relatively little data. The main takeaway from this analysis is that there appears to be very little connection.

These initial results already suggest that there is at most a weak connection between ice phenology and the solar cycle, but this rather simplistic method does not entirely answer the question, since (1) there could be stronger correlations when the sunspot data is lagged, and (2) we cannot immediately tell whether the few statistically significant correlations represent a real effect that occurs in a small number of locations or merely a result of the large sample size. The latter point could mean that the analysis has *overestimated* the impact of the solar cycle. On the other hand, in some of the locations a genuine connection could be masked by short time series, in which case the analysis could be *underestimating* the impact. Both possibilities deserve some attention.

We address the first issue in two different ways—first, by considering lagged correlations of ice phenology with sunspot data; and second, by checking Fourier transforms of the ice data alone to look for any 11-year periodicity. We address the second issue using two more methods—both of which involve studying the full spatial pattern of ice phenology rather than individual locations.

## Lagged correlations

To address the first question above, we check correlations for individual locations as before but with the sunspot data leading the ice data by 0 to 10 years in 1-year increments. (That is, we compare earlier sunspot data with later ice data.) In all 11 cases, the mean absolute value of the correlations for ice-on is about 0.11 or 0.12, and the number of statistically significant correlations is between two and five. For ice-off, the mean absolute values of lagged correlations are always between 0.09 and 0.14, with between zero and three being statistically significant. It is noteworthy that in both cases, the significant correlations are generally for short time series: when we change the minimum cutoff from 22 years to 33 years, almost no statistically significant correlations remain for any lag. The maps of lagged correlations between ice-on and the solar cycle (Fig 3) are again scattered and show little hint of spatial organization. The maps of lagged ice-off correlations with the solar cycle (Fig 4) do show some possible spatial organization, with clusters of locations of the same sign in Scandinavia, the North American Great

Ice-on correlation with sunspot number (lagged)

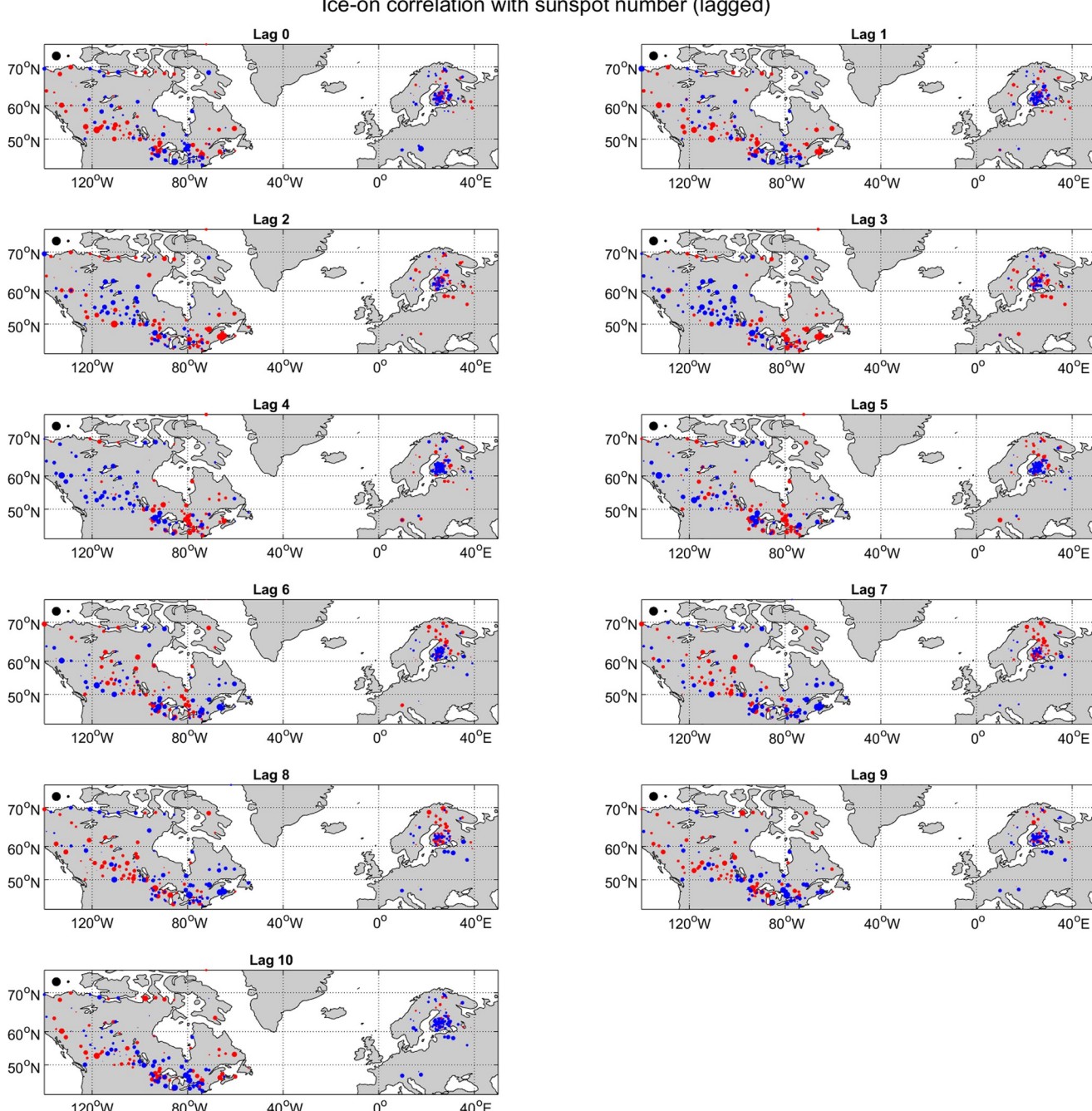

**Fig 3. Correlations between sunspot number averaged over the NDJ season and ice-on date.** Red and blue dots indicate positive and negative correlations, respectively, and the size of the dot indicates the absolute value of the correlation. The black dots in the upper-left corner represent correlations of 1 and 0.1, for scale. Note that statistical significance is not represented on this figure but is instead noted in the text.

Lakes region, and possibly western North America. Overall though, lagged correlations also do not show any strong evidence of a causal relationship between ice phenology and the solar cycle, with the exception of a possible spatial organization of lagged ice-off correlations. (We will follow-up on that possibility below).

Ice-off correlation with sunspot number (lagged)

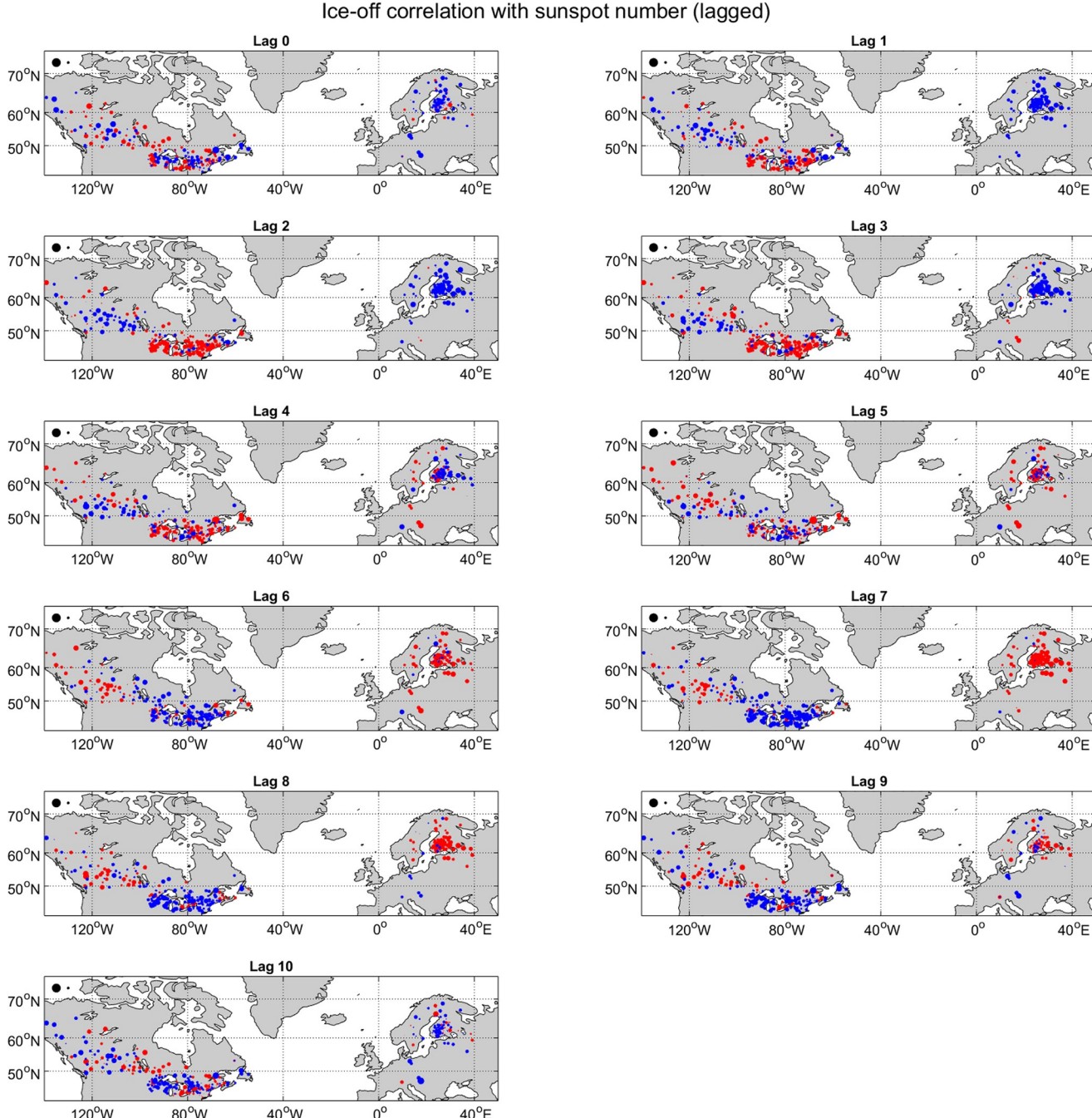

**Fig 4. As in Fig 2, but for correlations between sunspot number averaged over the MAM season and ice-off date.**

## Fourier transform

As a separate check, we use power spectra generated by the fast Fourier transform of the ice-on and ice-off timeseries to search for any 11-year periodicity regardless of the lag. An advantage of this method is that we can average the Fourier transforms over many locations without being concerned about whether the signs or lags differ by location.

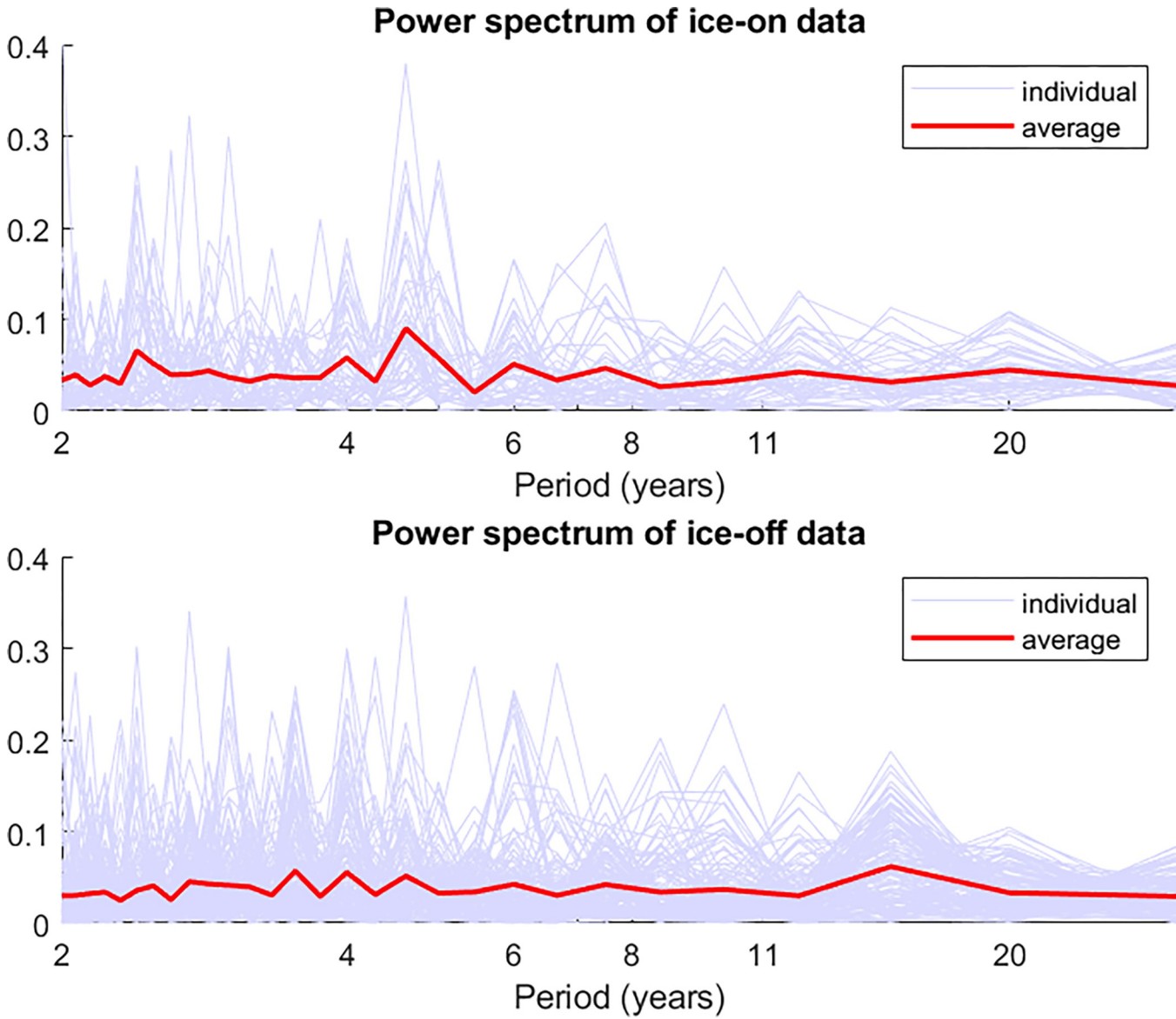

**Fig 5.** Power spectra (generated using fast Fourier transforms) of ice-on data (top) and ice-off data (bottom). Thin blue lines represent individual locations, and bold red lines represent the average. The spectra are calculated at all locations with at least 60 years of data. To allow easier comparison with the 11-year cycle we are looking for, we have transformed the curves so that the horizontal axis represents period, rather than the more standard frequency. The vertical axes have been normalized so that the area under each curve would be 1 if it was plotted on a linear horizontal axis.

For this analysis, we use only locations with at least 60 years of uninterrupted data (49 locations for ice-on and 142 for ice-off), and we use exactly 60 years of data for each such location, even if more is available. This allows us to avoid technical problems that arise with time series of differing lengths when we attempt to average the power spectra. Also note that while a Fourier transform is typically a function of frequency, we transform the curves so that the x-axis represents period instead.

Fig 5 shows the average power spectrum of ice-on dates (top) and ice-off dates (bottom) averaged over all locations. No clear 11-year periodicity is apparent in either individual locations (light blue) or in the averaged power spectra (bold red). This is in stark contrast to the power spectrum of the sunspot data itself (Fig 2).

## Spatial pattern

The second question posed above was whether the individual correlations (or lack thereof) were meaningful since (1) a few statistically significant correlations could occur coincidentally and (2) real relationships could be masked by the short time series in some locations. To address this, it will be helpful to consider not just individual locations, but the full dataset together. It is possible that a small impact of the solar cycle on lake/river ice that is difficult to discern at individual locations may be more clearly apparent in the aggregate data.

In order to check this, we determine whether the correlations between ice-on or ice-off and the sunspot cycle are spatially coherent. If, for example, we found that some region had predominantly positive correlations, and some other region predominantly negative, then even if the correlations were individually small and statistically insignificant, this overall spatial pattern might suggest the existence of a real effect.

We use Moran's I statistic as a measure of spatial autocorrelation. It is defined as:

$$I = \frac{\sum_{i=1}^{N} \sum_{j=1}^{N} w_{ij}(r_i - \bar{r})(r_j - \bar{r})}{\bar{w} \sum_{i=1}^{N} (r_i - \bar{r})^2}$$

where the $r_i$ and $r_j$ are the correlations between ice-off (or ice-on) and the solar cycle at the individual locations $i$ and $j$, $\bar{r}$ is the average of $r$, $N$ is the number of locations used, $w_{ij}$ is the weight assigned to the $(i, j)$ pair of locations, and $\bar{w}$ is the average weight [44]. This is the standard definition of the I statistic, but the weighting scheme may differ depending on the context. In some cases, $w_{ij}$ is taken to be 1 if the locations $i$ and $j$ are adjacent (in some suitably defined sense) and 0 otherwise. For this study, we instead define the weights with an exponentially decaying function of distance between the locations. Specifically,

$$w_{ij} = \exp\left(-\frac{\text{dist}(i,j)}{1000 \ km}\right)$$

where dist($i,j$) is the great-circle distance (shortest distance between two points on the surface of a sphere) from location $i$ to location $j$ in kilometers. This weighting scheme is designed to determine whether the individual correlations are themselves spatially autocorrelated over distances of roughly 1000 km and less.

Instead of attempting to interpret the numerical value of the I statistic directly, we use a bootstrapping method (or perhaps a Monte Carlo method, depending on one's definitions) to estimate its statistical significance—in other words, to ask whether the I value is larger than would be expected by chance. To do this, we randomize the temporal order of the sunspot data while leaving the ice data as it is, and then check the I statistic for each randomized time series. After performing this procedure 1000 times, the fraction of cases in which the randomly generated I statistic is greater than the original is then taken as an estimate of the p-value of the original.

We find that ice-on correlations with sunspot data have an I value that is not statistically significant for any lag. For ice-off, the unlagged p-value is not statistically significant ($p = 0.77$), but interestingly, this changes when we lag the sunspot data by two years ($p = 0.01$) suggesting a possible lagged correlation between sunspot number and ice-off. Some researchers have reported a possible 2-year lagged correlation between sunspot number and the NAO [45], though others have disputed this [46]. If such a relationship is real, it might explain the correlation seen here, given that the NAO is known to impact ice phenology (see Introduction). A few other lagged correlations are also statistically significant, as shown in Fig 6. This analysis agrees with the results of the previous section that the solar cycle has no detectable connection with ice-on, and a possible but weak connection with ice-off for some lag(s).

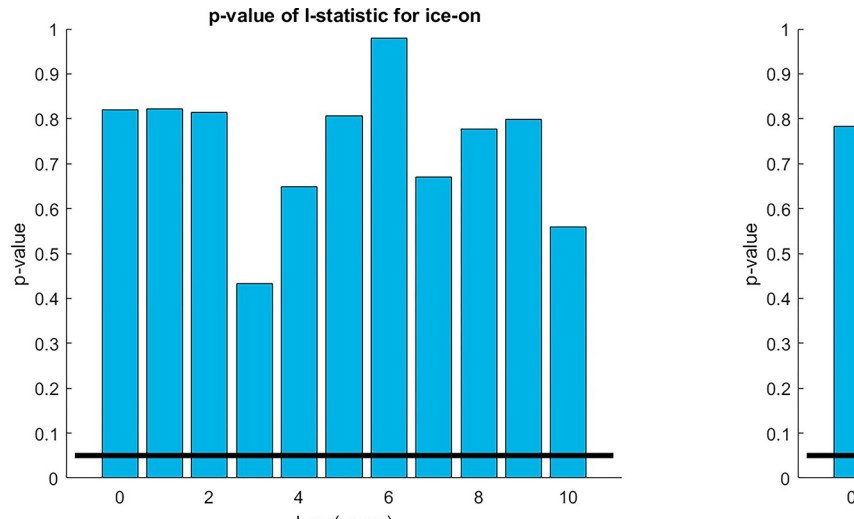
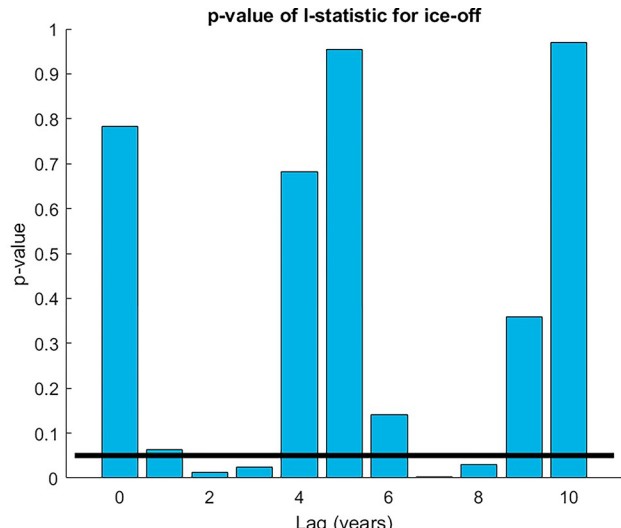

**Fig 6. The Moran's I statistic represents the spatial autocorrelation of the individual correlations.** Shown here are the p-values of the I statistic, estimated using a bootstrapping method described in the text. Values below the red line are statistically significant at the 0.05 level.

### Agreement between ice phenology and temperature

In the previous section we checked for spatial coherence of any kind in the correlations, without any preconceptions about what might be responsible for the connection between sunspot number and ice phenology. However, if this connection is mediated via temperature, then one could ask more specifically whether the lakes that melt early at solar maximum are also located in regions that are warmer at solar maximum, and vice versa. In other words, is the sign of the correlation between ice phenology and the solar cycle consistent with the sign of the correlation between local 2-meter air temperature and the solar cycle?

Specifically, we will consider the two kinds of correlations at some location to be "consistent" if the correlation of ice-on with the solar cycle and the correlation of temperature with the solar cycle are of the same sign (i.e. higher temperatures are associated with late freezing) or if the correlation between ice-off and the solar cycle has the *opposite* sign of the correlation between temperature and the solar cycle (i.e. higher temperatures are associated with early melting). We define a "percent consistent" number as the percent of lakes or rivers at which the correlations have consistent signs. Note that this simple metric considers only the signs of the correlations, and not their magnitudes.

In a completely random data set, one would expect the percent consistent number to be around 50%, but this would not likely be exact, so we need to ask how large this number would need to be before we would consider it meaningful. In other words, we again need a statistical significance test. We again use a bootstrapping method in which we keep the real temperature and lake ice data but randomize the temporal order of the sunspot data. We perform this procedure 1000 times. The fraction of cases in which the randomized percent consistent number is larger than the actual one is then taken as an estimate of the p-value.

We find that ice-on has a percent consistent number of 52% and an associated p-value of 0.28. Ice-off has a percent consistent number of 44% and a p-value of 0.99. Both of these are in contrast to the much larger percent consistent values found by [16] for the influence of the NAO (75%), PNA (80%), and ENSO (71%) on ice-off.

## Discussion

The analyses described above suggest that the solar cycle has no strong effect on lake and river ice phenology. Our *a priori* expectation from sensitivity analysis is that any effect of the solar cycle on surface temperature (and then on ice phenology) should be small. Correlations between ice-on (or ice-off) time series and the solar cycle calculated for each individual lake or river are almost never statistically significant. Lagged correlations show similarly weak results, and Fourier transforms of ice-on and ice-off time series do not show any clear 11-year periodicity. A visual inspection of the "Lag 0" panels Figs 3 and 4 suggests that the positive and negative correlations are scattered essentially randomly with no clear spatial pattern. Indeed, a test of spatial coherence of the correlations using Moran's I statistic produces generally weak results (with an exception discussed below). A somewhat more sophisticated test of spatial coherence compares (1) the spatial pattern of correlations between ice phenology and sunspot number as in Figs 3 and 4 with (2) the spatial pattern of temperature impacts of the solar cycle. This test again finds very weak and statistically insignificant results. Generally then, repeated analyses using a variety of methods converge on the conclusion that the solar cycle is not a strong driver of lake and river ice phenology.

The one interesting exception to the above story comes from lagged ice-off data. At lag 2 for example (that is, when comparing ice-off data with sunspot data from 2 years earlier) we find positive correlations clustered in eastern North America and negative correlations clustered in Scandinavia. The reverse occurs at lag 7. This is clear from a visual inspection of Fig 4, and the Moran's I statistic for these lags and a few nearby values is in fact statistically significant. Thus, there is the possibility of a weak delayed effect of the solar cycle on ice-off.

## Conclusions

Ice cover on lakes and rivers—and the timing of its freeze and melt cycle—is an important part of the physical environment of freshwater ecosystems. As such, it has attracted considerable research interest, and much of this research has focused on attempts to identify which of the well-known climate cycles (PDO, AMO, ENSO, PNA, NAO, etc.) play roles in controlling ice phenology. The 11-year sunspot cycle also has well-established impacts on Earth's atmosphere (though the strongest impacts are in the stratosphere) so it is worth considering in the same way. Due to its long frequency, if this cycle was at least partially responsible for modulating ice phenology, it would not only be of scientific interest from an earth-systems perspective but would also give a potential means of making long-term forecasts of ice phenology.

Our analyses described here using several different methods can rule out any strong influence of the solar cycle on ice phenology of lakes and rivers (though we cannot rule out the possibility of a weak lagged effect on ice-off specifically). The likely reason for this lies in the vertical distribution of the impacts of the solar cycle. Specifically, the amplitude of solar variability is much stronger in the ultraviolet than in the visible part of the spectrum. This ultraviolet light is mostly absorbed in the stratosphere, and accordingly the largest temperature changes associated with the solar cycle are found in the stratosphere. There are some surface-level temperature impacts as well, but they are considerably weaker. This result broadly agrees with previous research that has demonstrated that the impacts of the solar cycle on *surface-level* climate are generally quite weak, even if the middle atmosphere is more strongly affected.

In a different sense, this conclusion extends the results of previous work [16] that found that high-frequency oscillations such as the PNA, NAO, and ENSO which have periods of weeks or years generally have stronger impacts on ice phenology than low-frequency oscillations such as the PDO or AMO (and now the solar cycle) which have periods of decades. Ice

phenology therefore appears to be controlled largely by the sorts of short-term internal atmospheric variability that are impossible to predict in the long term.

The one major exception, of course, is that ice-covered seasons are becoming shorter as the global climate warms [7–18]. Hence, the one predictable aspect of ice phenology may simply be the by-now unsurprising prospect of shorter ice-covered seasons in the future. Year-to-year variations around that shorter mean duration, however, are likely to remain large and challenging to predict.

## Supporting information

**S1 Table. The number of locations with at least 20 years of data, at least 40 years of data, etc.**
(DOCX)

**S2 Table. The number of locations with ice-on data available in a given decade.**
(DOCX)

**S3 Table. The number of locations with ice-off data available in a given decade.**
(DOCX)

## Author Contributions

**Conceptualization:** Daniel F. Schmidt.

**Methodology:** Daniel F. Schmidt, Kevin M. Grise.

**Software:** Daniel F. Schmidt.

**Writing – original draft:** Daniel F. Schmidt, Kevin M. Grise, Michael L. Pace.

**Writing – review & editing:** Daniel F. Schmidt, Kevin M. Grise, Michael L. Pace.

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
