## [Decision Letter · Decision Letter 0]

3 Oct 2023

PONE-D-23-23784Does the 11-Year Solar Cycle Affect Lake and River Ice Phenology?PLOS ONE

Dear Dr. Schmidt,

Thank you for submitting your manuscript to PLOS ONE. After careful consideration, we feel that it has merit but does not fully meet PLOS ONE’s publication criteria as it currently stands. Therefore, we invite you to submit a revised version of the manuscript that addresses the points raised during the review process.

We look forward to receiving your revised manuscript.

Kind regards,

Rafael Duarte Coelho dos Santos, Ph.D.

Academic Editor

PLOS ONE

3. We note that Figures 2 and 3 in your submission contain [map/satellite] images which may be copyrighted. All PLOS content is published under the Creative Commons Attribution License (CC BY 4.0), which means that the manuscript, images, and Supporting Information files will be freely available online, and any third party is permitted to access, download, copy, distribute, and use these materials in any way, even commercially, with proper attribution. For these reasons, we cannot publish previously copyrighted maps or satellite images created using proprietary data, such as Google software (Google Maps, Street View, and Earth). For more information, see our copyright guidelines: http://journals.plos.org/plosone/s/licenses-and-copyright.

a. You may seek permission from the original copyright holder of Figures 2 and 3 to publish the content specifically under the CC BY 4.0 license. 

Additional Editor Comments:

The reviewers gave detailed instructions on what should be considered in the paper. Please try and answer as many of the concerns for another round of reviews.

Reviewers' comments:

Reviewer's Responses to Questions

**Comments to the Author**

1. Is the manuscript technically sound, and do the data support the conclusions?

Reviewer #1: Partly

Reviewer #2: Yes

Reviewer #3: Partly

2. Has the statistical analysis been performed appropriately and rigorously? 

Reviewer #1: No

Reviewer #2: Yes

Reviewer #3: Yes

3. Have the authors made all data underlying the findings in their manuscript fully available?

Reviewer #1: No

Reviewer #2: Yes

Reviewer #3: Yes

4. Is the manuscript presented in an intelligible fashion and written in standard English?

Reviewer #1: No

Reviewer #2: Yes

Reviewer #3: Yes

5. Review Comments to the Author

Reviewer #1: The manuscript predominantly relies on raw data, lacking necessary corrections or adjustments. One notable instance is the ice phenology data, which exhibits disparities in both spatial and temporal resolution. To ensure the credibility of this information, it necessitates validation through either on-site field observations or the integration of high-resolution remote sensing data. Similarly, the utilization of ECMWF data with a resolution one degree, with a lack of validation, raises concerns about its reliability for accurate assessment. Consequently, I find myself skeptical about the overall data quality as presented in its unprocessed state by the authors. Further efforts to refine and validate the data are imperative to bolster the manuscript's scientific rigor and integrity.

Reviewer #2: Potential link between the 11-year sunspot cycle and Lake and River Ice Phenology was investigated. The study showed no strong impact of the solar cycle and ice phenology. The topic is very interesting, and the text was written very well, but the figures suffer from low quality. My suggestions to improve this manuscript are listed below.

- No need to the given abbreviations in the Abstract.

- Line 107-110. Some these cited studies are suggested to be reviewed before talking about third possible driver of ice phenology.

- Introduction could be wider reviewing more recent studies. Some suggestions are:

** Pourderogar, H., et al. "Modeling and technical analysis of solar tracking system to find optimal angle for maximum power generation using MOPSO algorithm." Renewable Energy Research and Applications 1.2 (2020): 211-222

** Beiranvand, A., et al. "Energy, exergy, and economic analyses and optimization of solar organic Rankine cycle with multi-objective particle swarm algorithm." Renewable Energy Research and Applications 2.1 (2021): 9-23

** Zhou, J., Wang, L., Zhong, X., Yao, T., Qi, J., Wang, Y.,... Xue, Y. (2022). Quantifying the major drivers for the expanding lakes in the interior Tibetan Plateau. Science Bulletin, 67(5), 474-478. doi: https://doi.org/10.1016/j.scib.2021.11.010

- line 168: The way used to de-trend the values should be described.

- If the years in which a lake or river did not freeze included in your study, how your results may change?

- Figure 2 and 3 need legend and color bar.

- What kind of correlation was calculated? linear or nonlinear? justify. How about MI and lagged MI? see: Danandeh Mehr, A. (2023). A Gene-Random Forest Model for Meteorological Drought Prediction. Pure and Applied Geophysics, 1-11.

Reviewer #3: Using statistical methods such as the Fourier Transform and simple correlation coefficient, the authors investigate whether there is a connection between solar cycle and lake and river ice phenology over the Northern Hemisphere. However, I have several concerns that I would like to address:

- In addition to temperature, it's essential to consider the role of precipitation in ice cover dynamics. I recommend that the authors include precipitation data in their sensitivity analyses, alongside temperature.

- A dedicated Discussion section is necessary to contextualize the findings within the context of prior research. While some comparisons with previous work have been made, further discussion is warranted. It is recommended that the authors place a stronger emphasis on the key findings derived from the analyses conducted in this study. This would involve providing a more detailed and comprehensive discussion of the results obtained. By highlighting the specific outcomes and insights gained from their research, the authors can effectively underscore the significance of their work within the broader context of lake and river ice phenology and its potential relationship with solar cycle variations. This will help readers better understand the novel contributions of this study and its implications for our understanding of environmental and climatic dynamics.

- It would be interesting to explore the importance of the solar cycle on ice-off variability in comparison to other primary variables, such as temperature and precipitation. The authors should provide an explanation for the weak correlation between solar cycles and ice-off time series.

- In the Conclusions section (lines 386-399), the authors mention ocean-atmosphere circulations like PDO, NAO, AMO, ENSO, etc. as primary drivers of ice phenology, but the analysis and discussion related to these features and their correlation with ice variability are not presented in the manuscript. I also suggest they should improve the conclusions section by focusing on the findings obtained by the analyses made in the current work.

- For Figures 1 and 4, it's crucial to conduct significant analyses to determine the significance of the dominant periodicities that have been identified.

- For Figures 2 and 3, it's advisable to incorporate statistical tests to indicate where the correlations are statistically significant. One approach could be to mark points with significant correlations using cross-marks on the maps.

- Regarding Figure 4, it would be beneficial to display the upper and lower bounds of the power spectrum surrounding the averaged curve by considering the curves from all the ice recording points.

- In Figure 4, it would be helpful to provide additional explanations regarding the primary drivers of the observed dominant periodicities. Notably, there is a dominant mode at around 4.7 years in the ice-on data and approximately 15 years in the ice-off data. Exploring why these dominant modes differ and whether they have distinct drivers should be addressed.

- In Figure 4: Addressing the discrepancy in dominant patterns between ice formation (ice-on) and ice melting (ice-off) periods is crucial. It's worth considering whether distinct factors drive each of these phases, and if so, how it is possible for the primary influencing factors of ice-on and ice-off processes to vary, despite both occurring within the same hydrological system.

- The resolution of Figures 2 and 3 appears to be low and should be improved for better clarity.

- Including several examples of ice-on and ice-off time series would enhance the presentation of the data and help illustrate key points.

- Lines 217-218: The abbreviations NDJ and MAM should be defined to ensure clarity for readers.

- Lines 316-330 should be relocated to the methodology section for better organization and clarity.

6. PLOS authors have the option to publish the peer review history of their article (what does this mean?). If published, this will include your full peer review and any attached files.

Reviewer #1: No

Reviewer #2: No

Reviewer #3: No

---

## [Author Response · Author response to Decision Letter 0]

10 Oct 2023

Responses to Reviewer 1

The manuscript predominantly relies on raw data, lacking necessary corrections or adjustments. One notable instance is the ice phenology data, which exhibits disparities in both spatial and temporal resolution. To ensure the credibility of this information, it necessitates validation through either on-site field observations or the integration of high-resolution remote sensing data. Similarly, the utilization of ECMWF data with a resolution one degree, with a lack of validation, raises concerns about its reliability for accurate assessment. Consequently, I find myself skeptical about the overall data quality as presented in its unprocessed state by the authors. Further efforts to refine and validate the data are imperative to bolster the manuscript's scientific rigor and integrity.

We appreciate the reviewer’s concerns, and indeed, a study focused on the dynamics of individual lakes would require more attention to local detail than we have given here. Our focus, however, is different. We are considering the sunspot cycle as a potential driver of global climate variability. Hence, the signal we are looking for should be coherent across large spatial scales, and we need data from as many locations as possible in order to adequately study it. 

Thus, it is crucial that we have data spanning hundreds of locations over many decades (in some cases centuries). Under these circumstances, on-site field observations are clearly not plausible. However, it is also less important under these circumstances to have very high-resolution data and locally-specific detail, as the patterns we are looking for are much larger in scale. 

(With all that said, we have revisited the data set, and on closer examination, we did find one location error in the dataset which we have corrected and reported to the NSIDC.)

Responses to Reviewer 2

Reviewer #2: Potential link between the 11-year sunspot cycle and Lake and River Ice Phenology was investigated. The study showed no strong impact of the solar cycle and ice phenology. The topic is very interesting, and the text was written very well, but the figures suffer from low quality. My suggestions to improve this manuscript are listed below.

- No need to the given abbreviations in the Abstract.

We have removed the abbreviations. 

- Line 107-110. Some these cited studies are suggested to be reviewed before talking about third possible driver of ice phenology.

We have added some comments here. However, we keep these quite brief, as these other modes of variability are not the subject of the current study. 

- Introduction could be wider reviewing more recent studies. Some suggestions are:

** Pourderogar, H., et al. "Modeling and technical analysis of solar tracking system to find optimal angle for maximum power generation using MOPSO algorithm." Renewable Energy Research and Applications 1.2 (2020): 211-222

** Beiranvand, A., et al. "Energy, exergy, and economic analyses and optimization of solar organic Rankine cycle with multi-objective particle swarm algorithm." Renewable Energy Research and Applications 2.1 (2021): 9-23

** Zhou, J., Wang, L., Zhong, X., Yao, T., Qi, J., Wang, Y.,... Xue, Y. (2022). Quantifying the major drivers for the expanding lakes in the interior Tibetan Plateau. Science Bulletin, 67(5), 474-478. doi: https://doi.org/10.1016/j.scib.2021.11.010

The first two papers are about solar energy technologies and are not relevant to this manuscript, which is about the impact of the sunspot cycle on lake ice. The third paper is of more scientific interest, but it does not deal with ice, so again it is mostly unrelated to the topic of this manuscript. 

However, we have done an additional literature search and added some other new references to make sure the literature review is up to date. 

- line 168: The way used to de-trend the values should be described.

 We have added a note on the de-trending procedure.

- If the years in which a lake or river did not freeze included in your study, how your results may change?

This is an interesting question. Technically, in no-freeze years there simply is no ice-on or ice-off date to use in the calculations. One could perhaps treat the ice-on date as December 31 and the ice-off date as January 1 under such circumstances just to fill the gap, but that would still be somewhat arbitrary. 

This could be a real problem if it occurred frequently, but it is so rare in the NSIDC dataset (less than 1% of data) that it should be inconsequential to our conclusions. 

- Figure 2 and 3 need legend and color bar.

There are only two colors in those figures, so a colorbar cannot really be generated. However, we acknowledge that the original versions of these figures did come out at somewhat marginal resolution. We have re-generated these figures with improved quality. 

- What kind of correlation was calculated? linear or nonlinear? justify. How about MI and lagged MI? see: Danandeh Mehr, A. (2023). A Gene-Random Forest Model for Meteorological Drought Prediction. Pure and Applied Geophysics, 1-11.

The correlations referenced here are standard Pearson correlation coefficients. We have added a clarification to that effect. We do consider both lagged and unlagged correlations in considerable detail. 

Responses to Reviewer 3

Using statistical methods such as the Fourier Transform and simple correlation coefficient, the authors investigate whether there is a connection between solar cycle and lake and river ice phenology over the Northern Hemisphere. However, I have several concerns that I would like to address:

- In addition to temperature, it's essential to consider the role of precipitation in ice cover dynamics. I recommend that the authors include precipitation data in their sensitivity analyses, alongside temperature.

We thank the reviewer for this suggestion. We have repeated the sensitivity analysis for precipitation, and we find that the precipitation effect is considerably weaker than the temperature effect. We have added a paragraph to the text to describe this analysis. 

- A dedicated Discussion section is necessary to contextualize the findings within the context of prior research. While some comparisons with previous work have been made, further discussion is warranted. It is recommended that the authors place a stronger emphasis on the key findings derived from the analyses conducted in this study. This would involve providing a more detailed and comprehensive discussion of the results obtained. By highlighting the specific outcomes and insights gained from their research, the authors can effectively underscore the significance of their work within the broader context of lake and river ice phenology and its potential relationship with solar cycle variations. This will help readers better understand the novel contributions of this study and its implications for our understanding of environmental and climatic dynamics.

[WORK ON THIS]

- It would be interesting to explore the importance of the solar cycle on ice-off variability in comparison to other primary variables, such as temperature and precipitation. The authors should provide an explanation for the weak correlation between solar cycles and ice-off time series.

We have added some text to the conclusion to explain the likely reason for the weak connection between the solar cycle and ice phenology. 

- In the Conclusions section (lines 386-399), the authors mention ocean-atmosphere circulations like PDO, NAO, AMO, ENSO, etc. as primary drivers of ice phenology, but the analysis and discussion related to these features and their correlation with ice variability are not presented in the manuscript. I also suggest they should improve the conclusions section by focusing on the findings obtained by the analyses made in the current work.

The point of that statement was to contrast the drivers (like PNA, NAO, and ENSO) that do have significant impacts on ice-off with the subject of this study (the sunspot cycle) which does not. We have modified the text in an attempt to clarify that point. 

- For Figures 1 and 4, it's crucial to conduct significant analyses to determine the significance of the dominant periodicities that have been identified.

There are indeed methods for this, but in Figure 4, for example, the point is not to identify all periodicities in the data, but simply to note that there is no peak at 11 years. Since the 11-year peak does not exist, there is really nothing whose statistical significance we would need to test. 

- For Figures 2 and 3, it's advisable to incorporate statistical tests to indicate where the correlations are statistically significant. One approach could be to mark points with significant correlations using cross-marks on the maps.

Indeed, we would normally do this by using open circles and closed circles to distinguish between insignificant and significant correlations, respectively. However, in this case almost none of the correlations are significant (at least when considered individually), so instead we simply list the few correlations in the text. 

- Regarding Figure 4, it would be beneficial to display the upper and lower bounds of the power spectrum surrounding the averaged curve by considering the curves from all the ice recording points.

We appreciate the suggestion. We have replaced the figure with a new version that includes the individual curves as faint blue lines and the average as a bold red line. 

- In Figure 4, it would be helpful to provide additional explanations regarding the primary drivers of the observed dominant periodicities. Notably, there is a dominant mode at around 4.7 years in the ice-on data and approximately 15 years in the ice-off data. Exploring why these dominant modes differ and whether they have distinct drivers should be addressed.

The purpose of Figure 4 is simply to demonstrate that there is no clear 11-year peak in either power spectrum. Any other periodicities would be a subject for a different study. 

- In Figure 4: Addressing the discrepancy in dominant patterns between ice formation (ice-on) and ice melting (ice-off) periods is crucial. It's worth considering whether distinct factors drive each of these phases, and if so, how it is possible for the primary influencing factors of ice-on and ice-off processes to vary, despite both occurring within the same hydrological system.

Previous studies have indeed found that the drivers of ice-on and ice-off differ, essentially because of differing atmospheric conditions between fall and spring seasons. [add details and reverences]. The question of this study, however, is simply whether the solar cycle has a noticeable influence. Figure 4 does not show any such influence for either season. 

- The resolution of Figures 2 and 3 appears to be low and should be improved for better clarity.

Agreed. We have re-generated these images at considerably higher quality. 

- Including several examples of ice-on and ice-off time series would enhance the presentation of the data and help illustrate key points.

We appreciate the recommendation. We have added a new figure with a few representative time series. 

- Lines 217-218: The abbreviations NDJ and MAM should be defined to ensure clarity for readers.

These acronyms were defined in the previous section. They refer to the November-December-January (NDJ) season and the March-April-May (MAM) season. 

- Lines 316-330 should be relocated to the methodology section for better organization and clarity.

We appreciate the recommendation. We did indeed have some difficulty separating the “methods” and “results” sections of the paper, as many of the methods would not make sense until the reader sees the results from previous analyses. The resulting structure is a little different, but for this particular paper it seems to make the presentation flow more naturally.

---

## [Editor Report · Decision Letter 1]

14 Nov 2023

Does the 11-Year Solar Cycle Affect Lake and River Ice Phenology?

PONE-D-23-23784R1

Dear Dr. Schmidt,

We’re pleased to inform you that your manuscript has been judged scientifically suitable for publication and will be formally accepted for publication once it meets all outstanding technical requirements.

Kind regards,

Rafael Duarte Coelho dos Santos, Ph.D.

Academic Editor

PLOS ONE

Additional Editor Comments (optional):

I've read the response to the reviewers and am satisfied with it. I suggest the approval of the paper to the editors.
---

## [Editor Report · Acceptance letter]

20 Nov 2023

PONE-D-23-23784R1 

Does the 11-Year Solar Cycle Affect Lake and River Ice Phenology? 

Dear Dr. Schmidt:

I'm pleased to inform you that your manuscript has been deemed suitable for publication in PLOS ONE. Congratulations! Your manuscript is now with our production department. 

Kind regards, 

on behalf of

Dr. Rafael Duarte Coelho dos Santos 

Academic Editor

PLOS ONE